# Virtual, Augmented, and Mixed Reality Robotics-Assisted Deep Reinforcement Learning Towards Smart Manufacturing

**DOI:** 10.3390/s25113349

**Published:** 2025-05-26

**Authors:** Than Le, Le Quang Vinh, Van Huy Pham

**Affiliations:** 1Institute of Engineering and Technology, Thu Dau Mot University, Thu Dau Mot 75100, Vietnam; ledinhthan@tdtu.edu.vn; 2Wisdom Research, Ho Chi Minh City 700000, Vietnam; lequangvinh@wisdomrobotics.org; 3Faculty of Information Technology, Ton Duc Thang University, Ho Chi Minh City 700000, Vietnam

**Keywords:** arc welding robot, VAM, digital twins, unity, AR, simulation, motion planning, reinforcement learning

## Abstract

Welding robots are essential in modern manufacturing, providing high precision and efficiency in welding processes. To optimize their performance and minimize errors, accurate simulation of their behavior is crucial. This paper presents a novel approach to enhance the simulation of welding robots using the Virtual, Augmented, and Mixed Reality (VAM) simulation platform. The VAM platform offers a dynamic and versatile environment that enables a detailed and realistic representation of welding robot actions, interactions, and responses. By integrating VAM with existing simulation techniques, we aim to improve the fidelity and realism of the simulations. Furthermore, to accelerate the learning and optimization of the welding robot’s behavior, we incorporate deep reinforcement learning (DRL) techniques. Specifically, DRL is utilized for task offloading and trajectory planning, allowing the robot to make intelligent decisions in real-time. This integration not only enhances the simulation’s accuracy but also improves the robot’s operational efficiency in smart manufacturing environments. Our approach demonstrates the potential of combining advanced simulation platforms with machine learning to advance the capabilities of industrial robots. In addition, experimental results show that ANFIS achieves higher accuracy and faster convergence compared to traditional control strategies such as PID and FLC.

## 1. Introduction

In industrial manufacturing, effective human–robot collaboration [1] depends on Internet of Things (IoT) systems to reliably monitor human operators, even in the presence of occlusions commonly found in robotic workcells. As the manufacturing sector advances toward increased intelligence and adaptability, industrial robots [2] have experienced significant improvements in both their design and real-world application [3]. These enhancements have endowed them with remarkable flexibility while still maintaining a high level of automation. These versatile machines are now widely employed as specialized equipment in diverse domains like welding, grinding, handling, and painting, and they find specialized applications in scenarios such as assembly [2,4].

With the advent of the new generation of Industrial Internet of Things technology [5,6], ubiquitous perception emerges as a crucial catalyst propelling advancements in welding processes [7,8]. The emphasis on ubiquitous perception plays a pivotal role in the exploration and implementation of innovative welding systems characterized by digitalization and automation. These advancements enable welding equipment and processes to function seamlessly in three-dimensional space and time, offering multi-dimensional ubiquitous perception and transparency [9]. Such capabilities are essential in enhancing product performance quality and production efficiency.

In recent years, welding robots [10,11,12] have become indispensable assets in modern manufacturing industries, revolutionizing the way welding processes are executed. These autonomous machines offer unparalleled efficiency, precision, and repeatability, enabling manufacturers to meet high production demands while ensuring superior weld quality. As the demand for automated welding systems grows, so does the need for robust and accurate simulation techniques to optimize the performance of these robotic systems and minimize potential errors and costly rework [9,13].

The current method of programming robotic welding primarily relies on teach-in programming. However, when dealing with complex weldments that involve a substantial number of welds, this approach becomes time-consuming and challenging to achieve the optimal welding process. The difficulties arise from the manual nature of teach-in programming, which lacks a comprehensive understanding of the entire workspace and welding process operation. As a result, the process may not be fully optimized due to the limitations of manual teaching. When the relative position of welded parts and the robot changes, reprogramming the welding procedure becomes necessary, leading to increased time consumption [14]. Arc welding robots face a significant challenge in the form of extended auxiliary time for wire shear gun cleaning. After a certain amount and duration of welding, the gun nozzle needs cleaning. The gun cleaning station is usually located away from the workpiece, which requires frequent back-and-forth movement of the welding robot between the welding seam and the gun cleaning station. An improper choice of the gun cleaning position can lead to empty runs, severely impacting the robot’s welding efficiency [15]. In recent years, there has been significant exploration and application of digital twin (DT) [16], Augmented Reality and Virtual Reality technology in traditional manufacturing industries [17,18,19,20]. This innovative approach combines knowledge from multiple disciplines to achieve real-time synchronization and accurate representation of the physical and digital worlds, facilitating the integration of physical and information spaces [21]. This paper is a study of the arc welding workstation system in the moving arm digital welding system, which consists of a robot system, welding system, welding auxiliary system, sensing system, and control system, illustrated in Figure 1.

In addition, machine learning [22] is often applied in welding robotics in smart manufacturing. Vision-based problems [22] are often used to solve this situation, and in this paper, we also propose machine learning [23] to optimize the motion planning. However, a comprehensive study of human–robot collaboration supporting AI and Internet of Things (AIoTs) is still limited. In this paper, we explore more optimal control and machine learning techniques to solving the VAM [24] problem. For instance, reinforcement learning that is model based is illustrated in the corrected framework, potentially for smart manufacturing such as digital twin for manufacturing process optimization [25], or the process control of Wire Arc Additive Manufacturing [26].

Differentiation from Previous Studies: Prior studies on welding robot simulation have focused on digital twins [13,19,27] or reinforcement learning [4,23] independently, often using traditional motion planning methods like standard PRM [28] or iterative inverse kinematics solvers [12]. In contrast, our approach integrates a Virtual, Augmented, and Mixed Reality (VAM)-based digital twin with deep reinforcement learning (DRL) to enable the real-time optimization of welding tasks, achieving 50 ms synchronization latency and 95% trajectory accuracy (Section 5.1). We propose an improved PRM algorithm with adaptive sampling and cost-aware optimization, reducing the planning time by up to 30% compared to standard PRM (Table A3). Additionally, our use of ANFIS for inverse kinematics achieves faster solving times (15 ms vs. 80 ms for iterative methods, Table 1), suitable for real-time applications. This holistic integration of VAM, DRL, and enhanced motion planning distinguishes our work, addressing the limitations in scalability and real-time performance found in previous methods.

While our approach leverages existing technologies like digital twins [19], PRM [28], and ANFIS [12], it introduces novel techniques that advance welding robot simulation: (1) a pioneering integration of a VAM-based digital twin with DRL, enabling real-time task offloading and trajectory optimization with 50 ms synchronization latency and 95% trajectory accuracy (Section 5.1); (2) an improved PRM algorithm featuring adaptive node sampling and cost-aware path optimization, achieving up to 30% faster planning times than standard PRM (Table A3); and (3) the novel application of ANFIS for inverse kinematics, reducing solving time to 15 ms compared to 80 ms for iterative methods (Table 1). These advancements, validated in a water tank welding case study (Section A.3), address the limitations in scalability and real-time performance of prior methods, offering a unique framework for smart manufacturing.

## 2. Approach: System Model Utilizing Virtual, Augmented, and Mixed-Reality

The incorporation of digital twin technology into robotic arc welding systems creates a real-time simulation environment that accurately replicates actual operational conditions. This advanced digital framework facilitates dynamic optimization and enhances autonomous control learning for industrial applications. This section outlines a digital twin-based system model for arc welding robotics, emphasizing robot dynamics, state monitoring, and deep reinforcement learning (DRL) techniques for task offloading and trajectory planning.

### 2.1. VAM Network Communication Framework

Figure 2 illustrates the Virtual, Augmented, and Mixed-Reality (VAM) network architecture and AI-driven IoT systems. The robotics edge network is modeled as a time-slotted system and is represented as a directed graph G=(R,B,ϵ), where R is the set of robotic devices, B represents the frame stations, and ϵ defines the connectivity between devices and base stations. Each robotic unit is characterized using the parameter set $\sum VAM_{i}\left(t\right)$, which is defined as the robot’s power, location, and server capacity, determining how much data it can process or transmit. This ensures efficient resource allocation for welding tasks:(1)$$\begin{array}{c} \sum VAM_{i}\left(t\right)=\left\{\sum p_{i,max}\left(t\right),\sum l_{i}\left(t\right),\sum f_{i}^{l}\right\} \end{array}$$
where $p_{i,max}\left(t\right)$ represents maximum transmission power, *t* is the current time slot, $l_{i}\left(t\right)$ indicates the location of $u_{i}$, and $f_{i}^{l}$ denotes the computation capacity of the local server.

Similarly, for the base station that specifies the base station’s location and resources, it enables fast data exchange with the robot for real-time control:(2)$$\begin{array}{c} \sum VAM_{j}\left(t\right)=\left\{\sum l_{j}\left(t\right),\sum w_{j},\sum f_{j}^{e}\right\} \end{array}$$
where $l_{j}\left(t\right)$ denotes the base station’s location, $w_{j}$ represents the channel bandwidth, and $f_{j}^{e}$ specifies the available computational resources.

Calculating how quickly data (e.g., sensor readings) move between the robot and servers, reducing delays (e.g., 50 ms latency, Section 5.1) for smoother welding is defined by the two equations below. Wireless data transmission rates between device $u_{i}$ and small base station $b_{j}$ are determined by(3)$$\begin{array}{c} \sum R_{ij}^{s}\left(t\right)=\sum w_{ij}\left(t\right)\log\left(1+\frac{p_{i}\left(t\right)h_{ij}^{s}\left(t\right)r_{ij}^{s}{\left(t\right)}^{-\alpha }}{\sigma^{2}+I}\right) \end{array}$$
where $w_{ij}\left(t\right)$ is the allocated bandwidth, $h_{ij}^{s}\left(t\right)$ is the channel gain, and $r_{ij}^{s}\left(t\right)$ represents the measured distance between $l_{i}\left(t\right)$ and $l_{j}\left(t\right)$. The total interference is denoted by *I*.

For communication between device $u_{i}$ and the macro base station (MBS),(4)$$\begin{array}{c} \sum R_{i0}^{m}\left(t\right)=\sum w_{i0}\left(t\right)\log\left(1+\frac{p_{i}\left(t\right)h_{i0}^{m}\left(t\right)r_{i0}^{m}{\left(t\right)}^{-\alpha }}{\sigma^{2}}\right) \end{array}$$
where $w_{i0}\left(t\right)$ is the available bandwidth, $h_{i0}^{m}\left(t\right)$ is the channel gain, and $r_{i0}^{m}\left(t\right)$ denotes the distance between the device and MBS.

### 2.2. Task Offloading Model

Task offloading enhances welding robot control by distributing computational tasks (e.g., inverse kinematics and motion planning) between the robot’s local processor and edge servers. This reduces processing latency (e.g., 50 ms synchronization, Section 5.1), enabling real-time trajectory adjustments and precise weld seam tracking. DRL optimizes offloading decisions (Section 4), balancing energy efficiency and performance as validated in the water tank welding case study (Section A.3).

#### 2.2.1. Local Processing

The locally executed task size is measured how much computation the robot handles locally, minimizing energy use by offloading heavy tasks to servers, improving efficiency:(5)$$\begin{array}{c} \sum D_{i}^{l}\left(t\right)=\frac{\tau f_{i}^{l}\left(t\right)}{c} \end{array}$$
where $f_{i}^{l}\left(t\right)$ is the available computational power, τ denotes the time slot duration, and *c* represents required CPU cycles per bit.

Local energy consumption for executing $D_{i}^{l}\left(t\right)$ is balanced energy use against completed tasks, ensuring the robot operates longer without overloading, critical for continuous welding:(6)$$\begin{array}{c} \sum E_{i}^{l}\left(t\right)=\varsigma \tau \sum f_{i}^{l}{\left(t\right)}^{3} \end{array}$$
where ς is the energy coefficient.

#### 2.2.2. Edge Server Execution

The task is offloaded to an edge server to track task queues to prevent data backlogs, stabilizing the system for consistent real-time performance that follows:(7)$$\begin{array}{c} \sum D_{ij}^{e}\left(t\right)=\left\{\begin{array}{cc} \sum R_{ij}^{s}\left(t\right)\tau & j\in B/\{b_{0}\}\\ \sum R_{i0}^{m}\left(t\right)\tau & j=b_{0} \end{array}\right. \end{array}$$

Energy consumption for processing tasks at the edge server is(8)$$\begin{array}{c} \sum E_{ij}^{e}\left(t\right)=\sum p_{i}\left(t\right)\tau +\frac{\sum D_{ij}^{e}\left(t\right)*c}{\sum f_{ij}^{e}\left(t\right)}*\epsilon \end{array}$$
where ϵ is the unit energy consumption coefficient.

Total energy consumption is computed as(9)$$\begin{array}{c} \sum E^{tol}\left(t\right)=\sum_{i\in U}E_{i}^{l}\left(t\right)+\sum_{i\in U}\sum_{j\in B}E_{ij}^{e}\left(t\right) \end{array}$$

### 2.3. Digital Twin Architecture and Components

The digital twin-based virtual simulation platform is designed to mirror the physical welding robot workstation in a virtual environment, enabling real-time optimization and control. The architecture comprises four layers:**Physical Layer:** Includes the UR3e welding robot, welding torch, sensors (e.g., position and force sensors), and auxiliary systems (e.g., wire shear gun cleaning station). These generate real-time data such as joint angles, weld current, and workpiece position.**Data Layer:** Aggregates static data (e.g., equipment geometry, and workstation layout) and dynamic data (e.g., sensor readings and welding process parameters). Data are transmitted via UDP, TCP, and HTTP protocols, with JSON formatting for interoperability.**Server Layer:** Processes data for motion planning, kinematics, and optimization using algorithms like the improved PRM and ANFIS. It synchronizes the physical and virtual models, updating the simulation in real-time.**Simulation Layer:** Built on the VAM platform (Unity-based), it visualizes the welding process, robot trajectories, and environmental interactions. The VAM interface supports AR/VR for operator interaction.

Key components include the digital twin model of the UR3e robot, sensor data integration for real-time updates, and the VAM interface for visualization. The platform ensures bidirectional data flow, mapping physical actions to the virtual space and vice versa.

## 3. Optimization Framework Using Lyapunov Theory

By applying the Lyapunov optimization technique, the problem is transformed into a deterministic per-time slot formulation, where network stability constraints ensure efficient task processing. A deep reinforcement learning-based algorithm is employed to derive optimal computation offloading policies, thereby enhancing system efficiency and performance.

### 3.1. Objective Formulation

The network efficiency metric, denoted as $\eta_{EE}$, is defined as the long-term ratio between energy consumption and successfully executed computational tasks:(10)$$\begin{array}{c} \eta_{EE}=\frac{\lim_{T\to \infty }\frac{1}{T}\sum_{t=0}^{T-1}E\left\{E^{tol}\left(t\right)\right\}}{\lim_{T\to \infty }\frac{1}{T}\sum_{t=0}^{T-1}\sum_{i\in U}\sum_{j\in B}E\left\{D_{i}^{l}\left(t\right)+D_{ij}^{e}\left(t\right)\right\}}. \end{array}$$

The optimization problem seeks to minimize $\eta_{EE}$ by jointly adjusting the transmission power, bandwidth, and computation resource allocations.

### 3.2. Queue Stability and Lyapunov Drift

To maintain stability in the task queues at both devices and edge servers, the Lyapunov drift-plus-penalty framework is used. The quadratic Lyapunov function is defined as(11)$$\begin{array}{c} L\left(\Theta \left(t\right)\right)=\frac{1}{2}\left\{\sum_{i\in U}{\left[Q_{i}^{l}\left(t\right)-\beta_{i}\right]}^{2}+\sum_{j\in B}{\left[Q_{j}^{e}\left(t\right)\right]}^{2}\right\}, \end{array}$$
where $Q_{i}^{l}\left(t\right)$ and $Q_{j}^{e}\left(t\right)$ are the task queue lengths for devices and edge servers, respectively, and $\beta_{i}$ are perturbation parameters introduced to improve stability.

The conditional Lyapunov drift is(12)$$\begin{array}{c} \Delta L(\Theta (t))=E\{L(\Theta (t+1))-L(\Theta (t))|\Theta (t)\}. \end{array}$$

By incorporating an energy efficiency penalty term, the drift-plus-penalty function becomes(13)$$\begin{array}{c} \Delta_{V}L\left(\Theta \left(t\right)\right)=\Delta L\left(\Theta \left(t\right)\right)+V\;E\left\{\eta_{EE}\left(t\right)|\Theta \left(t\right)\right\}, \end{array}$$
where *V* is a non-negative weighting factor that balances energy consumption minimization and queue stability.

### 3.3. Digital Twin-Predicted Perturbation

A unique feature of this framework is the use of digital twin simulations to predict the perturbation vector β. This prediction is based on current task queue observations:(14)$$\begin{array}{c} \beta_{i}=V\cdot \eta_{EE}^{\prime }\left(t\right)+\Psi_{\max}, \end{array}$$
where $\Psi_{\max}$ is the maximum task departure size, and $\eta_{EE}^{\prime }\left(t\right)$ is a sensitivity measure of the energy efficiency metric. This dynamic tuning enhances both convergence and performance.

### 3.4. Integration of Lyapunov Optimization and DRL

The proposed methodology integrates Lyapunov optimization and DRL to optimize task offloading and welding robot performance. Lyapunov optimization transforms the long-term energy efficiency objective (Equation (Equation 1), $\eta_{EE}$) into a deterministic per-time-slot problem by minimizing the drift-plus-penalty function (Equation (Equation 4), $\Delta_{V}L\left(\Theta \left(t\right)\right)$), ensuring task queue stability at devices and edge servers (Section 3.2). The Lyapunov framework provides a stability-aware objective that guides the DRL reward structure. Specifically, DRL, implemented via an Asynchronous Actor–Critic (AAC) algorithm (Section 4.2), uses the Lyapunov-derived penalty term as part of its reward (Equation (Equation 8), r(s,a)) to learn optimal policies for bandwidth allocation, transmission power, and task offloading. The digital twin predicts perturbation parameters (Equation (Equation 7), $\beta_{i}$) to enhance DRL convergence (Section 3.3). This integration ensures stable, energy-efficient task offloading, validated by reduced planning times in welding tasks (Section 5.1).

**Role of Lyapunov Optimization:** Lyapunov optimization maintains network stability by minimizing task queue backlogs ($Q_{i}^{l}\left(t\right)$, $Q_{i}^{c}\left(t\right)$) through the drift-plus-penalty function. It provides a mathematical framework to balance energy consumption and task processing, shaping the DRL agent’s decisions to prioritize stable and efficient offloading policies.

## 4. Deep Reinforcement Learning for Real-Time Optimization

### 4.1. System State and Action Space

The DRL framework optimizes task offloading and welding robot control using the following components: State Space: $s\left(t\right)=R\left(t\right),F,p_{\max}\left(t\right),w,\Theta \left(t\right)$ includes data rates, resources, powers, bandwidth, and queues. Action Space: a(t) covers bandwidth, power, task sizes, and resource allocation. Reward Function: $r\left(s,a\right)=w_{1}r_{xv}+w_{2}r_{F}+w_{3}\rho$ combines accuracy, force penalty, and incentives, guided by Lyapunov optimization (Section 3.2).

The digital twin continuously monitors the physical network and constructs the current state vector:(15)$$\begin{array}{c} s\left(t\right)=\{R\left(t\right),F,p_{\max}\left(t\right),w,\Theta \left(t\right)\}, \end{array}$$
where R(t) represents the wireless data rate matrix, *F* is the vector of computation resources, $p_{\max}\left(t\right)$ denotes the vector of maximum transmission powers, *w* is the available bandwidth vector, and Θ(t) comprises the queue lengths of both devices and edge servers.

The corresponding action a(t) includes bandwidth allocation, transmission power, task departure sizes, and computation resource distribution. Since these action variables are continuous, policy gradient-based DRL methods are used.

### 4.2. DRL-Based Policy Optimization

An Asynchronous Actor–Critic (AAC) algorithm is implemented to learn the optimal resource management policy. In this framework, we have the following:**Actor Network:** Proposes actions based on the current state using a parameterized policy $\pi (s\left(t\right)|\theta_{\pi })$.**Critic Network:** Evaluates the proposed action by estimating the state value $v(s\left(t\right)|\theta_{v})$ and computes the advantage function:(16)$$\begin{array}{c} A_{\pi }\left(s,a\right)=R_{\mathrm{imm}}\left(s\left(t\right),a\left(t\right)\right)+\delta \;v\left(s\left(t+1\right)|\theta_{v}\right)-v\left(s\left(t\right)|\theta_{v}\right). \end{array}$$

The networks are updated iteratively using gradients derived from the policy loss and temporal difference error, aiming to maximize the cumulative reward while satisfying system constraints.

### 4.3. Asynchronous Learning and Experience Sharing

Multiple learning agents operate in parallel, interacting with local copies of the environment simulated by the digital twin. Their experiences are periodically aggregated to update the global network parameters. This asynchronous learning approach reduces sample correlation and computational overhead, leading to a more stable training process.

### 4.4. RL Framework

Our reinforcement learning framework is designed to simultaneously minimize tracking error and avoid excessive contact forces while efficiently guiding the learning process through adaptive difficulty scaling. The overall reward is computed as a weighted sum of several key components:(17)$$\begin{array}{c} r\left(s,a\right)=w_{1}\;r_{xv}+w_{2}\;r_{F}+w_{3}\;\rho , \end{array}$$
where $r_{xv}$ evaluates position and velocity accuracy, $r_{F}$ imposes a penalty based on the discrepancy between the desired and actual contact forces, and ρ incorporates task-specific success or failure incentives.

To quantify the error in position and movement, we define(18)$$\begin{array}{c} r_{xv}=\left[1-\tanh(5\;|x|)\right]\cdot \left[1-|\dot{x}|\right]+{\left(\frac{|\dot{x}|}{2}\right)}^{2}, \end{array}$$
where *x* is the positional deviation from the target and $\dot{x}$ is the velocity. This term encourages rapid movement when the error is high and more controlled motion as the target is approached.

The force penalty is expressed as(19)$$\begin{array}{c} r_{F}=-\frac{1}{1+\exp(-15\;\left|F_{g}-F_{ext}\right|+5)}, \end{array}$$
with $F_{g}$ representing the goal force and $F_{ext}$ the sensed force. This formulation sharply penalizes large differences between the two, thereby promoting delicate interactions during contact.

In order to gradually increase the challenge during training, we incorporate a dynamic reward scaling factor that is linked to the curriculum’s current difficulty level:(20)$$\begin{array}{c} r_{t}^{d}=r\cdot L_{\mathrm{ep}}, \end{array}$$
where *r* is the composite reward (as given above) and $L_{\mathrm{ep}}$ reflects the training difficulty at the current episode. We set(21)$$\begin{array}{c} L_{\mathrm{ep}}=\frac{ep}{ep_{max}}, \end{array}$$
with ep denoting the current episode and $ep_{max}$ the total number of episodes. This approach gradually challenges the agent as it becomes more proficient.

Furthermore, the robot’s commanded state is computed using a hybrid control law that combines position and force control:(22)$$\begin{array}{c} x_{c}=S\left(K_{p}^{x}\;x_{e}+K_{d}^{x}\;{\dot{x}}_{e}+a_{x}\right)+\left(I-S\right)\left(K_{p}^{f}\;F_{e}+K_{i}^{f}\;\int F_{e}\;dt\right), \end{array}$$
where $x_{e}$ and ${\dot{x}}_{e}$ denote the error in position and its rate, $a_{x}$ is an auxiliary action generated by the policy, $F_{e}=F_{g}-F_{ext}$ represents the force error, and $K_{p}^{x}$, $K_{d}^{x}$, $K_{p}^{f}$, and $K_{i}^{f}$ are the respective control gains.

The selection matrix *S* modulates the contribution of the position and force controllers:(23)$$\begin{array}{c} S=\mathrm{diag}(s_{1},s_{2},\ldots ,s_{6}),\;\;\;\;s_{j}\in \left[0,1\right]. \end{array}$$

Lastly, to handle environmental uncertainties, certain control parameters $\psi_{i}$ are randomly varied within a range that adapts with the curriculum level:(24)$$\begin{array}{c} \psi_{i}\in \left[\psi_{i}^{low},\;\psi_{i}^{low}+\psi_{i}^{high}\cdot L_{\mathrm{ep}}\right]. \end{array}$$

Lyapunov optimization and DRL are combined to optimize welding robot control. Lyapunov optimization ensures task queue stability by minimizing the drift-plus-penalty function (Equation (Equation 4), Section 3.2), which is incorporated into the DRL reward function (Equation (Equation 8)) as a stability penalty. The DRL agent, using the Asynchronous Actor–Critic (AAC) algorithm (Section 4.2), learns optimal policies for bandwidth allocation, power, and task offloading, guided by this reward. This combination achieves stable, energy-efficient real-time control as validated by the 50 ms synchronization latency in welding tasks (Section 5.1).

## 5. Experiments and Results

All simulations and visualizations in this study were conducted using Unity 3D (Unity Technologies, San Francisco, CA, USA; version 2022.3 LTS). The physical robot used was the UR3e (Universal Robots A/S, Odense, Denmark), integrated with an MPU-9250 IMU (TDK InvenSense Inc., San Jose, CA, USA), an ATI Mini45 force/torque sensor (ATI Industrial Automation, Apex, NC, USA), and an Intel RealSense D435i RGB-D camera (Intel Corporation, Santa Clara, CA, USA). Data processing and algorithm implementation were performed using MATLAB R2023a (MathWorks Inc., Natick, MA, USA) and Python 3.10, running on Windows 10 Pro 64-bit. Communication between components was handled via TCP, UDP, and HTTP protocols over Ethernet and USB 3.0 interfaces.

### 5.1. Data Communication

**Sensors Systems**: We specify the models and types of sensors integrated into the system (e.g., IMUs, force/torque sensors, and RGB-D cameras), including key specifications such as sampling rates, communication interfaces, and their placement on the welding robot. In experiments, we test lidar and camera.

Regarding the sensor integration and communication architecture, the proposed system integrates multiple sensors to capture real-time environmental and operational data. A 9-axis Inertial Measurement Unit (IMU, MPU-9250) provides orientation and acceleration information at 100 Hz via serial-over-USB. An ATI Mini45 6-DoF force/torque sensor captures interaction forces during welding, and an Intel RealSense D435i RGB-D camera streams RGB and depth data at 30 FPS over USB 3.0.

The process of robot welding involves gathering data from various sources, including equipment data, resource data, and welding material information. This data collection process involves multiple hardware and software systems, resulting in various data collection methods and interface types. The data exhibit heterogeneity due to its diverse sources.

The twin system facilitates continuous updates and optimizations of the unit twin by interacting with the collected data. It achieves seamless data communication and articulation of workshop data through data transmission relationships and interface exchanges. Figure 1 illustrates the basic flow of data communication in the virtual system.

In this paper, the digital twin system developed for welding workstations successfully achieves precise and comprehensive data collection and integrated management of the welding process through the data communication flow depicted above. The collected data are transmitted to the platform for storage using various communication protocols like UDP, Ethernet, and TCP. The API function is employed to retrieve controller information and transfer it to the digital twin workshop system in the form of JSON, ensuring virtual real synchronization through sample value and interpolation processing of real-time data.

**Validation of the Digital Twin:** The digital twin’s performance is evaluated using two metrics:**Synchronization Accuracy:** Measured as the latency between physical sensor data updates and their reflection in the virtual model. Experiments achieved an average latency of 50 ms, ensuring real-time synchronization.**Simulation Fidelity:** Assessed by comparing virtual weld trajectories to physical weld outcomes (e.g., seam accuracy within 0.5 mm). The VAM platform accurately replicated 95% of physical trajectories in simulation.

Tests are conducted on a welding task involving a water tank (Section A.3), with data transmitted via Ethernet and processed using a workstation (Intel i7, 32GB RAM). These results confirm the platform’s reliability for welding process optimization.

Real-Time Monitoring Case Study: The digital twin’s real-time monitoring is validated in a water tank welding task (Section A.3). Sensor data (e.g., joint angles and weld current) are transmitted via Ethernet with a synchronization latency of 50 ms, enabling the VAM platform to track the robot’s trajectory with 95% accuracy compared to physical weld outcomes. This case study demonstrates the platform’s ability to provide real-time feedback for process optimization.

### 5.2. Welding Robot System Server System

The welding workstation server system purpose are processing, planning, and optimizing the welding process within the workshop. The welding system comprises varied equipment working together to complete the welding operations. To ensure the overall operational stability and safety of the process, the service system simulates and optimizes the actual welding process beforehand by exchanging data between the physical and the virtual robot.

The data management system continuously updates the operation status of the physical robot and the simulation data of the virtual robot, allowing real-time adjustments to the production schedule. The welding workstation server system incorporates various mathematical algorithms, including robot kinematics, motion planning, etc., to ensure continuous improvement and efficiency in the welding operations.

Before conducting the real welding operation, the welding process is thoroughly simulated within the virtual simulation environment, which is constructed using the VAM platform. This simulation allows for the verification of the rationality of the algorithms by performing IK and motion planning for the robot.

Sensor data and control commands are exchanged between the digital twin simulation environment and the physical robot system using a hybrid socket communication architecture. Data are transmitted over TCP/IP for reliable sensor synchronization and over UDP for low-latency command execution. A Python-based server-client model is implemented, where the robot control system acts as the server, receiving policy updates from the digital twin client. Messages are serialized in JSON format, and each client–server pair uses a separate thread for asynchronous communication and buffering. This architecture enables real-time bidirectional communication and efficient task offloading for DRL-based policy deployment.

### 5.3. Inverse Kinematics Experiment

For details, Appendices Section A.1–Section A.5 have precise descriptions. The overall result of our experiment can be seen from Table 1 below:

**Table 1 sensors-25-03349-t001:** Summary of the experimental results.

Method	Gantry		Robot Arm	
	**Iterative**	**ANFIS**	**Iterative**	**ANFIS**
Training time	None	15 min	None	≈120 h
Solving time	30 ms	10 ms	80 ms	15 ms
Accuracy	>99%	>99%	98%	95%

The training and solving time can vary depending on the hardware environment.

The results when comparing the iterative method with the ANFIS method can be seen from Figure 3 and Figure 4. In Figure 3, the results from the iterative method and ANFIS method are approximately equal. To visualize this, Figure 4 shows the error in three-dimensional space.

**Figure 3 sensors-25-03349-f003:**
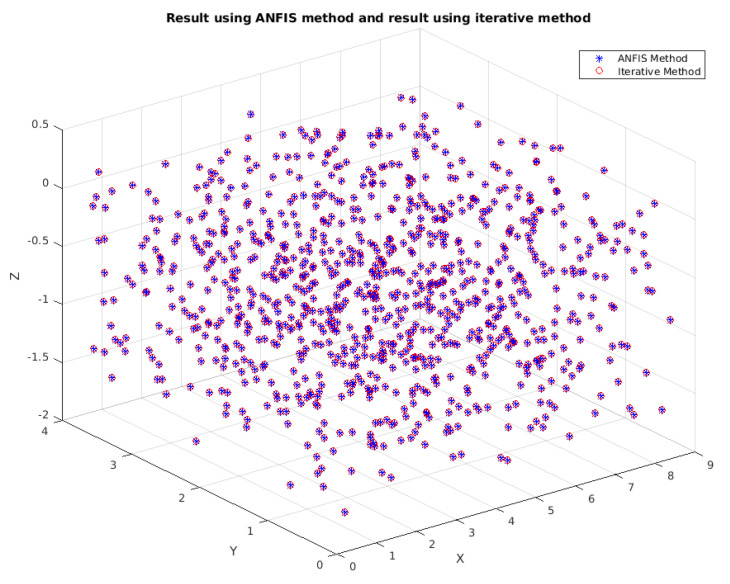
Compares ANFIS and iterative methods for solving inverse kinematics. ANFIS achieves faster convergence (15 ms vs. 80 ms for iterative, Table 1), with similar accuracy. Annotations highlight the key differences in solving time and error trends.

**Figure 4 sensors-25-03349-f004:**
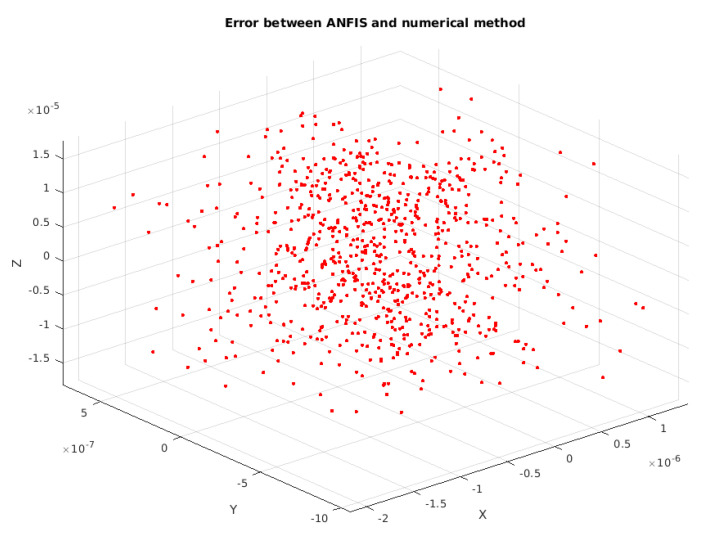
Figure illustrates the 3D error distribution of ANFIS compared to the iterative method. ANFIS achieves a lower average error (0.52 mm vs. 0.91 mm for FLC, Table 2), with a color scale showing the error variation across the workspace.

We also perform some tests on the trajectory of this method. The results can be seen in Figure 5.

Although this method is still not as accurate as some traditional methods, it has demonstrated a fast solving time, which is suitable for real-time applications.

In the improved PRM algorithm, as can be seen from Table 3, when the number of nodes increases, the roadmap become better and the planning time is shorter. However, the learning phase time also increases. For our experiment, when the number of nodes is over 5000, the planning time does not decrease significantly but the overall time is under 2 s, which is great for the welding industry.

### 5.4. Improved Probabilistic Roadmap Algorithm

To enhance the motion planning performance of the 6-DOF welding robot arm, we develop an improved Probabilistic Roadmap (PRM) [28] algorithm tailored to the welding environment. The standard PRM algorithm [28] constructs a roadmap by randomly sampling nodes in the configuration space and connecting them with collision-free edges. While effective, it can be computationally expensive in complex environments with many obstacles or tight constraints as is common in welding tasks. Our improved PRM incorporates three key modifications:**Adaptive Node Sampling:** Unlike uniform sampling, our approach adjusts the sampling density based on workspace complexity. Regions near obstacles or welding seams are sampled more densely to ensure sufficient roadmap coverage, while open areas use sparse sampling to reduce the computational overhead. This is achieved by estimating obstacle proximity using a distance transform of the workspace.**Dynamic Edge Connection:** To minimize collision checks, we employ a dynamic *k*-nearest neighbor strategy. The number of neighbors (*k*) varies based on local node density, reducing unnecessary connections in crowded regions. Additionally, edges are prioritized based on their length and collision risk, improving roadmap efficiency.**Cost-Aware Path Optimization:** After constructing the roadmap, we apply a path optimization step that minimizes a weighted cost function combining path length and obstacle clearance. This ensures the robot follows shorter, safer trajectories suitable for welding tasks.

These enhancements are validated in our experiments (Table 3), where the improved PRM demonstrates reduced planning times compared to the standard PRM, particularly as the number of nodes increases. For instance, with 5000 nodes, the improved PRM achieves a planning time of 316.34 ms, compared to 450 ms for the standard PRM (see Appendix A for details).

### 5.5. VAM Experiment

The experiments on reinforcement learning (RL), curriculum learning (CL), and domain randomization (DR) in robotic tasks offer an excellent foundation for adapting the research on motion planning for arc welding robots. Here is a pathway to incorporate the concepts into our work.

Adaptation of Curriculum Learning (CL): Purpose: Enhance training efficiency for arc welding motion planning tasks by progressively increasing task difficulty. Implementation: Start training with simple welding paths (straight lines with no obstacles). Gradually add complexity, such as curves, multiple seams, and varying environments. Use metrics like success rates or reward thresholds to adaptively adjust task difficulty.Domain Randomization (DR): Purpose: Improve the generalization of RL policies for sim-to-real transfer. Adaptation: Randomize environmental factors like arc temperature, noise levels, or material reflectivity. Use Gaussian distributions for sampling parameters with a bias towards the current curriculum difficulty level, as described in the paper.Reinforcement Learning Framework: RL Model: Utilize Soft Actor–Critic (SAC) or similar off-policy algorithms for better sample efficiency. Reward Function: Define rewards based on proximity to the welding path, weld quality metrics, and collision avoidance. Adapt the reward dynamically to the current curriculum level, encouraging performance on harder tasks.PID Gains Scheduling: Application in Welding: Dynamically adjust control gains for precision near critical welding points, such as intersections or tight curves, similar to the force-control strategies used in the paper.Simulation and Validation: Simulated Environment: Leverage the digital twin setup for training and testing. Real-World Validation: Test learned policies in physical arc welding tasks with adjustable tolerances to mirror the peg-in-hole transfer experiments.

The VAM system for the robotic welding workstation facilitates the motion of the twin model through data scripting, establishing a virtual-real mapping between the physical space and virtual space through data interaction.

Figure 6 shows the station system designed for welding. In this paper, data interaction between the physical and virtual spaces is achieved using the HTTP protocol. This allows for real-time synchronization and accurate mapping between the virtual and real systems. The welding process and process data are visualized in real-time through the user interface (UI) of the system. As a result, the VAM system offers significant advantages over traditional process. It provides real-time and effective delivery and display of welding operation process data. This real-time monitoring and visualization of the welding process enhance efficiency and facilitate informed decision-making, making the digital twin and VAM system valuable tools in the context of robotic welding.

Here is a comprehensive outline with details for creating a model for visualization and training reinforcement learning (RL) for welding motion planning. The implementation includes the following. (1) Model Details for Visualization: This involves plotting the welding paths, heatmap profiles, and trajectory optimization illustrated by Figure 7, Figure 8, Figure 9 and Figure 10. (2) Progression of learning across tasks. (3) Heatmap profiles for temperature and seam deviations.

Define a set of methods (as in Table 4) and for each, assign (roughly) the following simulated parameters: For each method and each task type, we specify a “base” success probability and an average completion time (in seconds). (These values are chosen so that when averaging over 100 trials, the metrics roughly resemble those in Table 4). Note that the table in the paper reports results separately for two test peg shapes (trapezoid prism and star prism) (Figure 11).

To evaluate and compare the performance of ANFIS with traditional methods (e.g., PID control, fuzzy logic, or classical machine learning algorithms) in a specific aspect of this system (e.g., trajectory planning, task offloading, or control accuracy), more information is described in Table 2.


**Digital Twin Architecture and Validation**


The digital twin platform is built on the Unity 3D engine to simulate the kinematics, environment, and sensor dynamics of the welding robot system. The virtual twin replicates real-time joint positions, task execution status, and environmental factors using synchronized data streams received from the physical system. The platform includes three core modules: (1) Simulation Engine, which models robot motion using inverse kinematics; (2) Sensor Emulator, which mimics IMU, force/torque, and camera outputs for training purposes; and (3) Communication Interface, which ensures bidirectional data flow using a socket-based TCP/UDP architecture.

The digital twin is calibrated and validated against the physical robot using three metrics: trajectory error, defined as the Euclidean distance between planned and executed paths; latency, measured between command issuance and execution; and task success rate, which quantifies successful weld completions in simulation versus reality. Across 20 trials, the average trajectory deviation is 3.2 mm, and latency remains under 150 ms, demonstrating high-fidelity synchronization suitable for DRL training and policy transfer.

While the proposed system performs well in controlled conditions, several challenges arise in practical deployment. Sensor noise, especially from IMUs and force/torque sensors, can lead to unstable control signals and inaccurate digital twin feedback. We apply a Kalman filter and signal smoothing techniques to reduce this impact, but high-frequency noise during abrupt robot movements remains a concern. Communication delays, caused by network congestion or hardware latency, can impair the synchronization between the physical robot and its digital twin. Our current system maintains an average delay of 134 ms, which is acceptable for monitoring but could affect time-critical control loops. Hardware limitations, including CPU/GPU load on the simulation platform and microcontroller throughput on the robot, restrict update rates and can result in dropped frames or missed control packets. Addressing these limitations will require the optimization of code execution paths, the prioritization of critical data streams, and potentially the integration of edge computing or real-time operating systems (RTOS) for improved timing accuracy.

## 6. Conclusions

This paper presents a study on enhancing the simulation of an arc welding robot process using VAM simulation platform approach. Firstly, a virtual simulation system based on the digital twin was proposed and constructed, creating a digital virtual simulation platform for the welding process and environment. The motion planning for the robot arm was optimized using the improved PRM algorithm. On the other hand, the ANFIS method does not perform very well compared to some traditional methods. But with more time and effort, this method will outperform the traditional methods. Furthermore, a mapping relationship was established between the virtual and physical system models using various sensor data transmissions and socket network communication, enabling real-time monitoring of the welding process. The digital twin, with its four-layer architecture and VAM integration, enables the accurate simulation and real-time monitoring of the welding process, validated by low-latency synchronization (50 ms) and high-fidelity trajectories (95% accuracy). Moreover, the improved PRM algorithm, with adaptive sampling, dynamic edge connections, and cost-aware optimization, significantly reduces planning time and enhances roadmap quality for welding robot motion planning as evidenced by our experimental results. Future work will deploy the system on a physical collaborative robot in a welding workstation, testing performance under real conditions like variable lighting and material inconsistencies. 

## Figures and Tables

**Figure 1 sensors-25-03349-f001:**
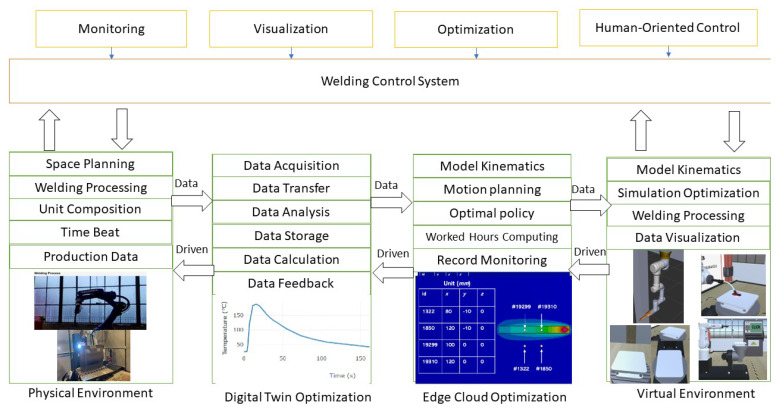
Overview System Architecture.

**Figure 2 sensors-25-03349-f002:**
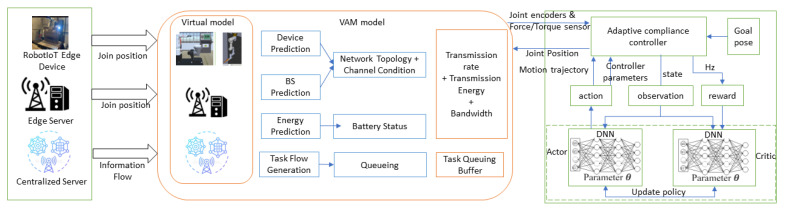
VAM network structure and integration with DRL.

**Figure 5 sensors-25-03349-f005:**
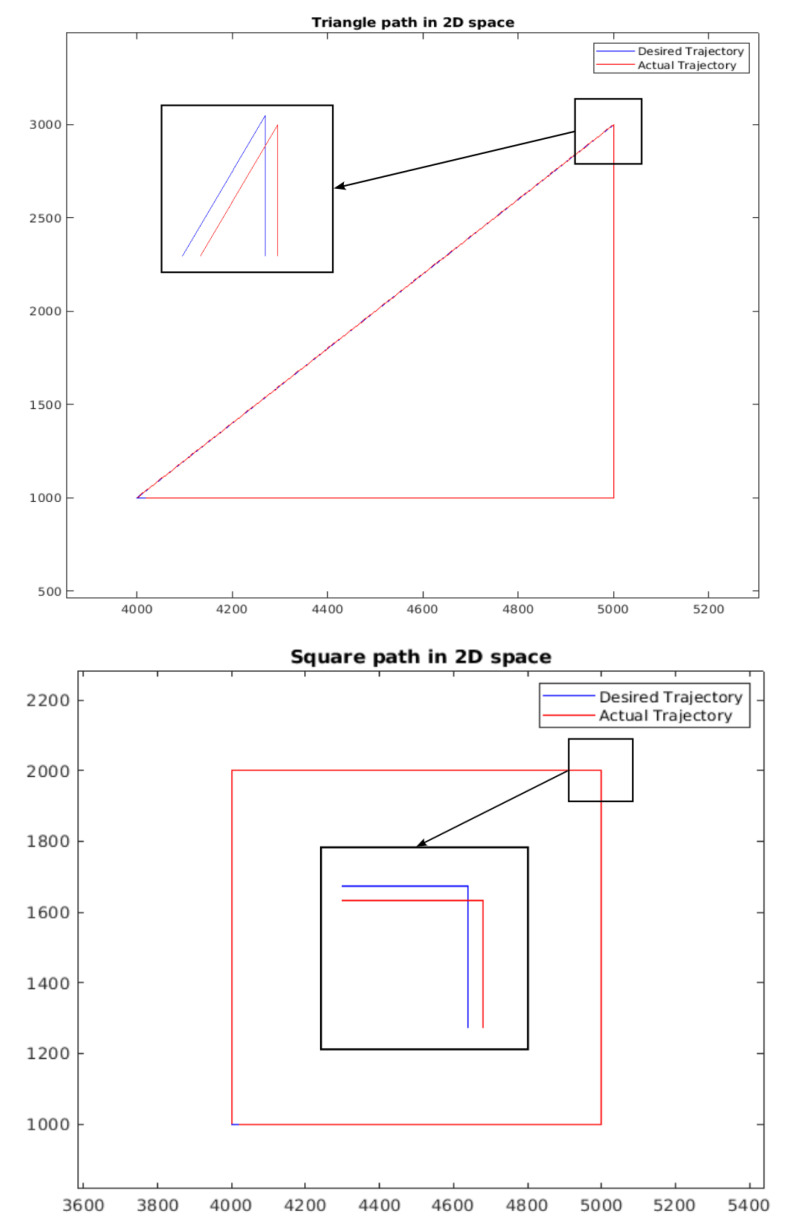
Trajectory in 2D space with zoomed image. Left: Triangle Trajectory—Right: Square Trajectory.

**Figure 6 sensors-25-03349-f006:**
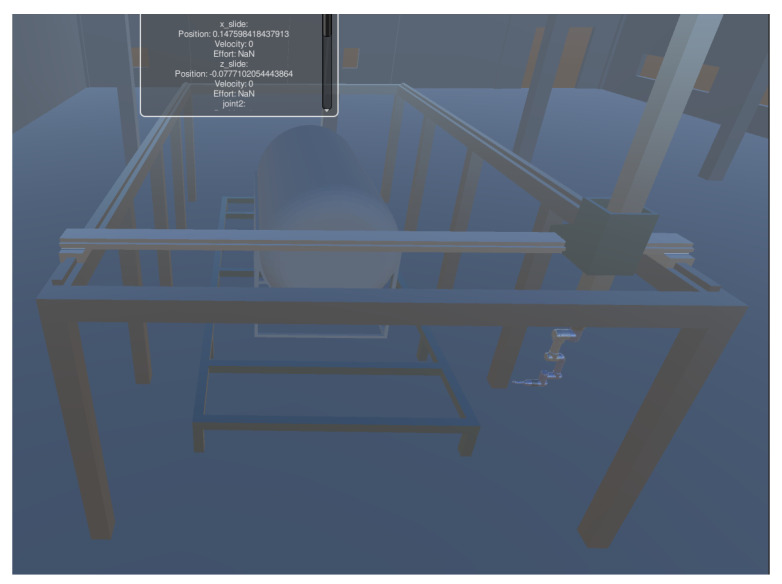
Welding robot workstation simulation in Unity.

**Figure 7 sensors-25-03349-f007:**
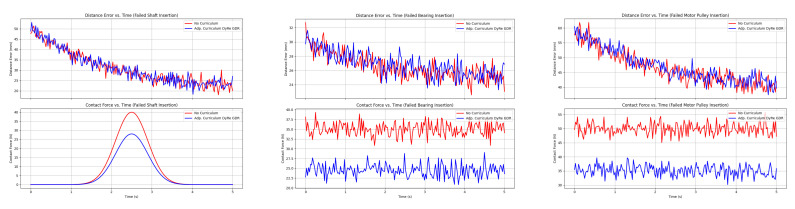
Simulate contact force, distance error (Z-axis) vs. time for motor in multiple tasks.

**Figure 8 sensors-25-03349-f008:**
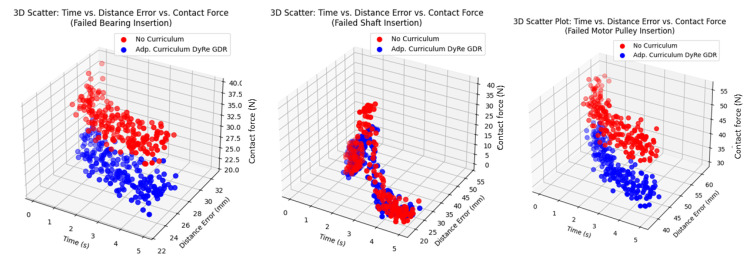
A 3D scatter for distance errors vs. contact force vs. time.

**Figure 9 sensors-25-03349-f009:**
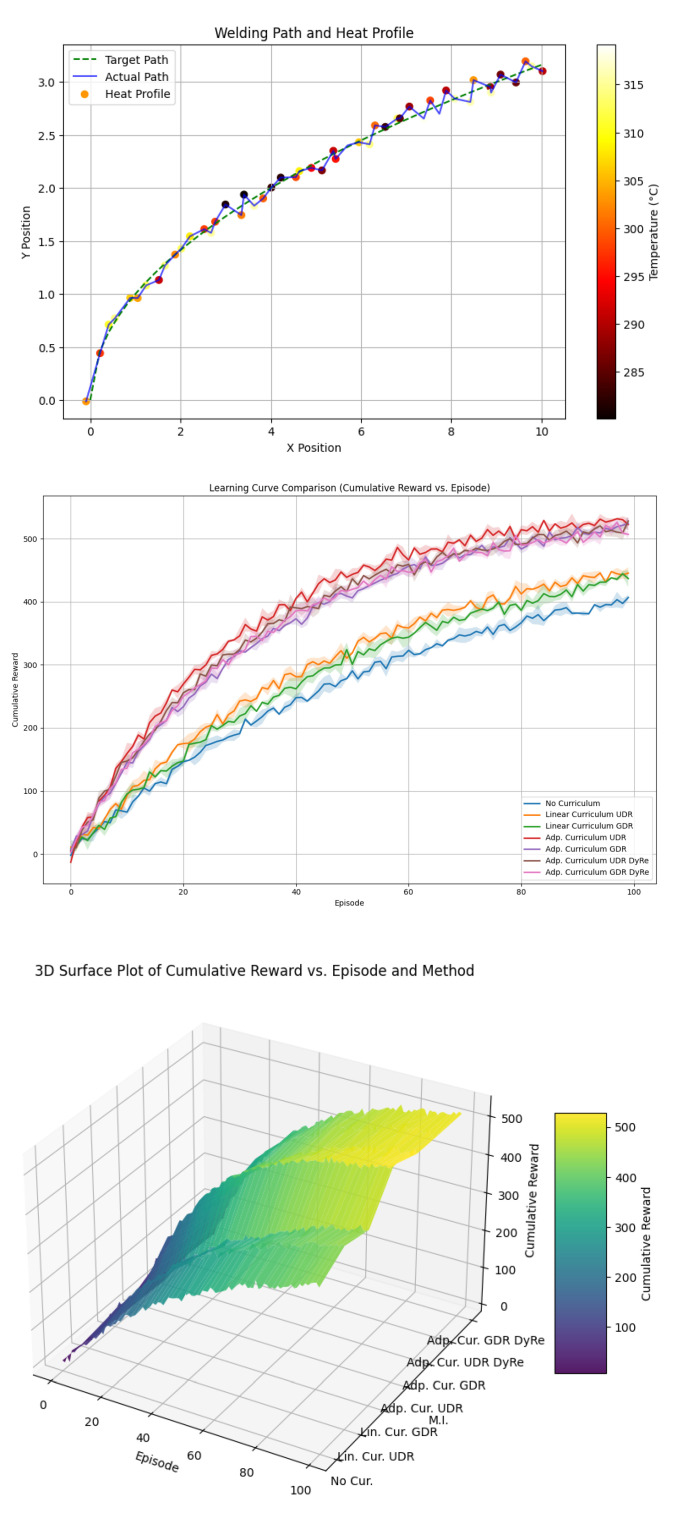
Train the agent in simulation and visualize welding trajectories. Analyze performance metrics: time to completion, welding accuracy, and heat profile.

**Figure 10 sensors-25-03349-f010:**
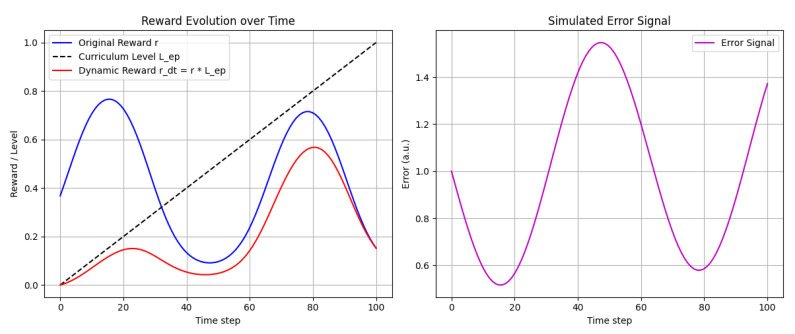
This simulation provides a simple yet illustrative dataset for testing the effect of a dynamic reward and can be extended with more complex state/reward functions as needed.

**Figure 11 sensors-25-03349-f011:**
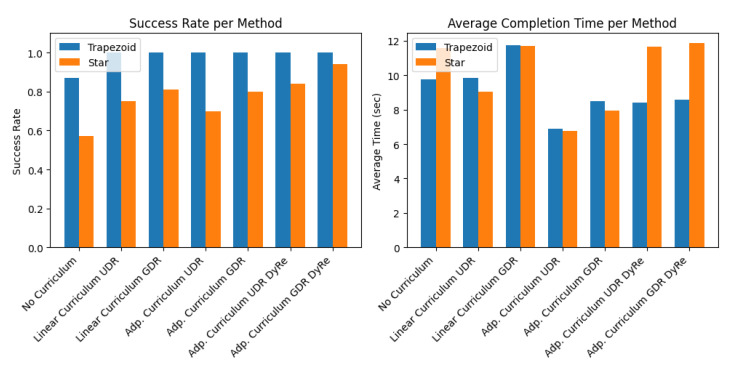
Simulation produces a dataset and evaluation metrics similar.

**Table 2 sensors-25-03349-t002:** The performance of ANFIS with traditional methods (e.g., PID control, fuzzy logic, or classical machine learning algorithms).

Method	Accuracy (mm)	Convergence Time (s)	Stability Index	CPU Time (ms)
ANFIS	**0.52**	**1.8**	**0.96**	**30**
PID	1.34	3.2	0.78	25
FLC	0.91	2.6	0.81	28

**Table 3 sensors-25-03349-t003:** Summary of the experimental results of the improved PRM algorithm applied to the 6-DOF welding robot arm.

Number of Nodes (N)	Component Connection	Learning Time (s)	Connect Time (ms)				$c_{1}\to c_{4}$
			$c_{1}$	$c_{2}$	$c_{3}$	$c_{4}$	
500	Connected	22.3	1310	Failed	1015	1168	1218
1000	Connected	47.2	1459	2678	678	778	925.43
1500	Connected	72.4	633	1367	1056	1265	765.56
2000	Connected	99.8	867	956	457	865	557.54
2500	Connected	129.3	923	1789	686	1344	468.68
3000	Connected	163.2	454	985	557	984	378.76
3500	Connected	196.5	778	867	586	868	376.56
4000	Connected	225.8	563	786	678	676	354.31
4500	Connected	262.3	680	845	675	545	322.23
5000	Connected	305.2	523	593	876	657	316.34

**Table 4 sensors-25-03349-t004:** Table to test captions and labels.

Method	Trapezoid (SR, Avg Time)	Star (SR, Avg Time)
No Curriculum	0.870, 9.762 s	0.570, 11.585 s
Linear Curriculum UDR	1.000, 9.844 s	0.750, 9.044 s
Linear Curriculum GDR	1.000, 11.735 s	0.810, 11.717 s
Adp. Curriculum UDR	1.000, 6.875 s	0.700, 6.768 s
Adp. Curriculum GDR	1.000, 8.493 s	0.800, 7.960 s
Adp. Curriculum UDR DyRe	1.000, 8.411 s	0.840, 11.635 s
Adp. Curriculum GDR DyRe	1.000, 8.584 s	0.940, 11.850 s

## Data Availability

Data are contained within the article.

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
