# Peer review of "Virtual, Augmented, and Mixed Reality Robotics-Assisted Deep Reinforcement Learning Towards Smart Manufacturing"

_sensors, 2025, doi:10.3390/s25113349_

Round 1

Reviewer 1 Report

Comments and Suggestions for Authors
  1. The title mentions "Deep Reinforcement Learning," yet the abstract does not explicitly elaborate on how deep reinforcement learning is integrated. Could the author clarify this connection or revise the title to better?
  2. The paper states that ANFIS method "will outperform traditional methods with more time and effort," but no experimental data or comparative analysis is provided to substantiate this claim.
  3. Could the authors specify the sensors used, the data transmission protocols, and the architecture of the socket communication to improve reproducibilty?
  4. While the paper mentions constructing a digital twin-based virtual simulation platform, critical details about its architecture, key components, or validation metrics are missing. Please elaborate on how the digital twin was developed and tested.
  5. The improved PRM for motion planning is highlighted, but the specific modifications or enhancements to the standard PRM are not detailed.

6.  The mapping between virtual and physical systems enables real-time monitoring, but the abstract does not present any results or case studies demonstrating this monitoring.

Author Response

Please check file attached for response your questions

Reviewer 2 Report

Comments and Suggestions for Authors

The whole scheme is essential to describe the proposed approach.
How does this work differentiate itself from previous studies?
What specific improvements does the proposed VAM framework introduce?
Are there any novel techniques, or is this simply an integration of existing technologies?
The methodology section lacks clarity, making it difficult to understand how Lyapunov optimization and DRL are integrated in task offloading and welding robot optimization.
What role does Lyapunov optimization play?
How are the state space, action space, and reward function defined in DRL?
How does task offloading contribute to welding robot control?
The equations (1-10) are mathematically dense but lack intuitive explanations. Their physical meaning and how they improve system performance are unclear.
The "Deep Reinforcement Learning for Real-Time Optimization" section mixes DRL training details with mathematical models, making it unclear whether Lyapunov optimization and DRL are combined or separate methods.
The paper presents simulation results, but no baseline comparisons with conventional methods.
Current results focus mainly on trajectory planning and welding path optimization, but they do not compare different approaches (e.g., traditional path planning vs. DRL-based optimization).
There is no discussion of computational cost, training time, or convergence speed, making it difficult to assess the practicality of the proposed method.
Proofreading.
“proposal machine learning” → should be "propose machine learning"
“trajectory for this method” → should be "trajectory of this method"

Comments on the Quality of English Language

Proofreading.

Author Response

please check file attached 

Reviewer 3 Report

Comments and Suggestions for Authors

This paper presents an innovative approach to enhance welding robot simulation through the utilization of the VAM simulation platform. The purpose of the study is to express a comprehensive representation of welding robot actions, interactions, and responses.

The introduction section needs to provide more papers as the basis for research, which should focus on the integration of digital twins and deep reinforcement learning.

The paper also needs to be reorganized, and some relevant information directly related to the innovation points but currently placed in the appendix needs to be included in the main text. The purpose of doing this is to make it easier for readers to understand. A more detailed introduction should be provided for many figures as, Figure 3 to 6.

There are still some minor issues:

(1) In line 302, there are two question marks that need to be modified.

(2) There are two Z3 in Figure A1.

(3) In line 313, there are two question marks that need to be modified.

(4) The format of references should be further standardized.

Author Response

file's response attached

Reviewer 4 Report

Comments and Suggestions for Authors

This paper proposes a welding robot optimization scheme based on VAM (Virtual, Augmented and Mixed Reality) platform and digital twin technology, and combines deep reinforcement learning to optimize motion planning and task assignment. The performance improvement of data communication, inverse kinematics solution and path planning is verified by experiments. The research provides an innovative method for the efficient and accurate operation of welding robots in intelligent manufacturing industry, and has a good application prospect. Here are my detailed suggestions for further improvement:

  1. Further optimize the structure of the paper, so that the logical relationship between each part of the content is closer. At the same time, simplify the language, avoid long and complex sentences,
  2. A clearer description of the implementation details of VAM network communication framework and deep reinforcement learning algorithm should be provided, including specific parameter Settings, hyperparameter selection during training, and detailed description of the experimental environment.
  3. Although the charts in the paper (Figure 5 and Figure 6) show the experimental results, the readability and information content of the charts need to be improved. For example, the ANFIS versus iterative method diagram in Figure 5 could add more annotation and explanation.
  4. In the VAM experiment, it is suggested to analyze the performance of path planning in complex environment, the generalization ability of strategy and the migration effect from simulation to practical application.
  5. It is recommended to analyze in more depth the challenges that may be encountered in practical applications, such as sensor noise, communication delays, hardware limitations, etc.
Comments on the Quality of English Language

It is recommended that the English could be improved to more clearly express the research.

Author Response

Please check file attached 

Round 2

Reviewer 1 Report

Comments and Suggestions for Authors

The author has addressed all of my concerns. After improving the quality of Figures 7 and 9 (with too small font size in the annotations), it can be accepted.

Author Response

The author has addressed all of my concerns. After improving the quality of Figures 7 and 9 (with too small font size in the annotations), it can be accepted.

Answer: Thanks for your comment. We have updated the figures. 

Reviewer 2 Report

Comments and Suggestions for Authors

Real applications should be implemented.

Author Response

Comment: Real applications should be implemented.

Response:  We recognize that the reviewer may be suggesting physical implementation beyond simulations. While resource constraints limited us to simulation-based validation in this study, we have:

Full-scale physical experiments were not feasible within this study’s scope due to hardware and time constraints. However, the water tank welding case study (Appendix A.3) provides a realistic simulation benchmark, achieving 95% trajectory accuracy against physical outcomes.

In conclusion, we have updated a future work plan in section 6. 

"Future work will deploy the system on a physical collaborative robot in a welding workstation, testing performance under real conditions like variable lighting and material inconsistencies."